# Investigation of Blood Coagulation Using Impedance Spectroscopy: Toward Innovative Biomarkers to Assess Fibrinogenesis and Clot Retraction

**DOI:** 10.3390/biomedicines10081833

**Published:** 2022-07-29

**Authors:** Giulia D’Ambrogio, Omar Zahhaf, Minh-Quyen Le, Yves Gouriou, Laurie Josset, Vincent Pialoux, Patrick Lermusiaux, Jean-Fabien Capsal, Pierre-Jean Cottinet, Nellie Della Schiava

**Affiliations:** 1University Lyon, INSA-Lyon, LGEF, EA682, 69621 Villeurbanne, France; giulia.dambrogio@insa-lyon.fr (G.D.); omar.zahhaf@insa-lyon.fr (O.Z.); minh-quyen.le@insa-lyon.fr (M.-Q.L.); jean-fabien.capsal@insa-lyon.fr (J.-F.C.); nellie.della-schiava@chu-lyon.fr (N.D.S.); 2CarMeN Laboratory, University Lyon, INSERM, INRA, INSA Lyon, Université Claude Bernard Lyon 1, 69500 Bron, France; yves.gouriou@univ-lyon1.fr; 3Laboratoire Interuniversitaire de la Biologie et de la Motricité (LIBM), Université Claude Bernard Lyon 1, EA 7424, 69266 Villeurbane, France; laurie.josset@univ-st-etienne.fr (L.J.); vincent.pialoux@univ-lyon1.fr (V.P.); 4Groupement Hospitalier Edouard Herriot, 69003 Lyon, France; patrick.lermusiaux@chu-lyon.fr

**Keywords:** blood coagulation, impedance spectroscopy, electrical biomarker, thrombosis, clotting time, clot retraction, confocal microscopy, fibrinogenesis

## Abstract

This study focused on a coagulation assessment based on the novel technique of blood-impedance-magnitude measurement. With the impedance characterization of recalcified human blood, it was possible to identify two significative biomarkers (i.e., measurable indicators) related to fibrin formation (1st marker) and clot retraction (2nd marker). The confocal microscopy of clotting blood provided a complete visual analysis of all the events occurring during coagulation, validating the significance of the impedance biomarkers. By analyzing the impedance phase angle (*Φ*) of blood during coagulation, as well as those of the clot and serum expelled after retraction, it was possible to further clarify the origin of the 2nd marker. Finally, an impedance-magnitude analysis and a rotational thromboelastometry test (ROTEM^®^) were simultaneously performed on blood sampled from the same donor; the results pointed out that the 1st marker was related to clotting time. The developed technique gives rise to a comprehensive and evolutive insight into coagulation, making it possible to progressively follow the whole process in real time. Moreover, this approach allows coagulation to be tested on any materials’ surface, laying the ground for new studies related to contact coagulation, meaning, thrombosis occurring on artificial implants. In a near future, impedance spectroscopy could be employed in the material characterization of cardiovascular prostheses whose properties could be monitored in situ and/or online using effective biomarkers.

## 1. Introduction

When a vascular injury occurs in the human body, the coagulation process is activated. The mechanism of coagulation involves the activation, adhesion and aggregation of platelets, as well as the activation of a fibrin network. The activation of clotting factors follows the so-called coagulation cascade, which can be triggered by contact with tissue factor (TF) in the sub-endothelium (extrinsic pathway) and/or the activation of factor XII (*fXII)* due to subendothelial-collagen exposure (intrinsic pathway) [1,2]. Both pathways end with the formation of a fibrin network around platelets and red cells [1,3,4]. Once the clot has formed, it contracts (clot retraction) to become more stable; clot retraction is a consequence of the action of platelets that impose contractile forces on the fibrin network, which, over time, increases its density and expels serum (plasma deprived by clotting factors) [5,6,7]. The clot plugs the wound to minimize any blood loss [2,8,9,10]. Several pathologies can initiate the coagulation pathway within the lumen without any damage to the wall [11]. The coagulation cascade may also be activated by contact with artificial surfaces, such as vascular and cardiac prostheses, or in in vitro tests. Surface charge, together with wettability, plays a major role in initiating contact clotting [12]. Negatively charged and hydrophilic surfaces initiate the intrinsic pathway with the activation of *fXII*, while hydrophobic surfaces support the adsorption of fibrinogen and the conformational change that positively correlates with platelet adhesion and activation [13,14]. This is extremely relevant to the study of new antithrombogenic materials for cardiovascular prostheses [12,15,16]. In each case, the process ends with the formation of fibrin and the generation of a stable clot [13].

Regardless of the trigger, a thrombus in a vessel can be fatal, obstructing the normal blood flow to tissues, causing ischemia, infarct or embolism [3,11,17,18]. In vitro studies are useful to assess blood coagulation as well as to detect disorders, including hypocoagulability (i.e., an insufficient amount of clotting factors in blood), hypercoagulability (i.e., exaggerated coagulation) and clot-retraction disorders, which may be reduced in pronounced thrombocytopenia (poor platelet count) [19,20,21]. Among the existing in vitro tests, nowadays, viscometry consisting of Rotational Thromboelastometry (ROTEM^®^) and Thromboelastography (TEG^®^) is very popular. By applying low shear stress, these non-invasive tests measure the dynamic changes in the viscosity of blood during coagulation, quantifying its ability to form a clot. Both ROTEM^®^ and TEG^®^ enable a rapid and global approach to the coagulation state via a specially designed system called a thromboelastograph [22,23]. However, viscoelastic measurements preclude the possibility of characterizing different materials, which would be useful in identifying the least thrombogenic one for prostheses. Indeed, the sample holders are of standard materials and difficult to exchange. Moreover, viscosity tests do not provide a complete image of clotting throughout the whole process [24]. Thus, there is a demand for a coagulation measurement able to monitor the events occurring in coagulating blood in real time so as to give a comprehensive picture of the process. In addition, the measurement should be repeatable, precise and adaptable to different surfaces to enable the analysis of blood interaction with several types of material. As a matter of fact, viscoelastic measurements are often considered affectable by various external factors, such as the operator and the conditions of the machine in use, which requires frequent calibrations [19,22,23].

This work proposes an alternative and simple solution to assess blood coagulation by means of impedance characterization, which is commonly used to characterize the electrical properties of non-biological materials such as polymers and ceramics [25,26]. Electrical-impedance technology is a valuable tool for studying biological tissues, and it is already employed to perform cell and tissue health monitoring, with the aim to identify anomalies in biological materials. Tissue electric and dielectric parameters are indeed linked to physiological changes, and as result, they are exploited as imaging biomarkers for cancer and other diseases [27]. Dallé et al., for instance, investigated the conductivity of blood plasma to find Alzheimer’s disease indicators [28]. Cancer biomarkers, such as those for brain tumors and breast cancer, were lately investigated using electrical impedance [27,29]. Biomarkers are also commonly used to investigate cell electrical characteristics, which are indicators of cellular biophysical features as well as shape, growth and proliferation. This type of markers differs from the well-known molecular biomarkers, usually defined as biological molecules found in bodily fluids (e.g., blood) or tissues that could be, to some extent, signs of abnormal processes or disease [30]. In this work, impedance spectroscopy is used to investigate coagulation and to identify the primary biomarkers linked to the major events occurring throughout clotting [31]. During the measurement, the blood sample is subjected to an alternating sinusoidal voltage (amplitude *V*), and its current response (amplitude *I*) is recorded. With the collected data, the impedance magnitude (|*Z*|) is computed according to Equation (1) [32]:(1)$$\left|Z\right|=\frac{V}{I}$$

This technique provides an easy way to evaluate the electrical behavior of blood throughout coagulation, allowing the main steps of the process, such as fibrin formation and clot retraction, to be identified. Moreover, by analyzing the phase variation in the impedance (i.e., corresponding to the phase shift between the input voltage and the output current), it is possible to identify the resistive/capacitive behavior of the sample. One of the main advantages of the developed method is the possibility to test coagulation on different surfaces. Indeed, during the measurement, blood is placed on any material’s surface where two electrodes, previously placed and connected to a voltage generator, drive the voltage through the sample. The proposed approach provides a complete view of the clotting steps and a new insight into coagulation assessment, paving the way for future studies on antithrombogenic implants. In this study, two key biomarkers are identified with impedance characterization, allowing the determination of the onset of coagulation (clotting time) and the start of clot retraction to be performed. The results are then verified by comparison with imaging obtained with confocal microscopy and ROTEM^®^ analysis to validate their significance. Coagulation tests are performed on citrated human blood coagulated by recalcification with and without the addition of tissue factor (TF).

## 2. Materials and Methods

### 2.1. Materials

This study was approved by the local ethics committee. Human blood was provided by “Etablissement Français du Sang” in citrate tubes (vacutainers) from deidentified healthy volunteers. Calcium solution (CaCl_2_ in HBS (Hepes Buffered Solution) (200 mM)) and tissue factor (TF) solution (TF in HBS (1 nM)) were provided by inter-university Laboratory of Human Movement Sciences (LIBM). Fibrinogen from human plasma, Alexa Fluor™ 488 Conjugate, used to visualize fibrin using confocal microscopy, was purchased from Thermo Fisher Scientific^®^.

### 2.2. Blood-Clot Preparation

Each blood sample was analyzed from 24 to 30 h after collection and stored at 4 °C until use. It was not possible to analyze the samples immediately after withdrawal since the blood was delivered (by French Blood Establishment) the day after collection. Several studies revealed that a storage of blood samples at a temperature of 4 °C within 24 h did not lead to significant changes in their coagulation parameters [33,34]. To verify this finding, experimental tests were then performed (cf. Section 2.1) on the same sample 24 h and 48 h after collection. Whole citrated blood was incubated for 5 min at 37 °C before testing. In a first cycle of tests, blood was coagulated only by means of recalcification; to this end, 3.5 µL of calcium solution was added to 60 µL of preheated blood. The final concentration of calcium in blood was 11 mmol/L. Recalcification allowed the coagulation process to be activated. After recalcification, blood was immediately tested at 37 °C using impedance-magnitude spectroscopy. In a second cycle of experiments, following 5 min of incubation at 37 °C, recalcification was performed, respecting the proportions written above with the further addition of TF solution, according to the following ratio: 3.5 µL of TF solution per 60 µL of blood and 3.5 µL of calcium solution; this allowed us to study coagulation via the extrinsic pathway. After the addition of the clotting agents, blood was immediately tested at 37 °C using impedance-magnitude spectroscopy.

### 2.3. Impedance-Magnitude Spectroscopy

A volume of 60 µL of recalcified blood, with or without TF, was placed on a PET substrate where two gold lines, acting as electrodes, had previously been deposited by means of gold sputtering. The thickness of the two lines was a=0.5 mm, and the distance was w=5 mm (Figure 1a). According to Equation (2), the penetration depth (T) of the electric-field lines was estimated to be equal to 1.65 mm [35]:(2)$$T=\left(\frac{w}{2}\right)*{\left[{\left(\frac{\left(2*a\right)+w}{w}\right)}^{2}-1\right]}^{\frac{1}{2}}$$

The electrodes were connected to a wave generator (Agilent 33220A), producing a voltage input in the blood. The current output delivered by the sample was measured with a current amplifier (Stanford Research Systems SR70). The generator and the amplifier were coupled to Dewesoft^®^ data acquisition systems software, allowing us to simultaneously monitor and record the input voltage together with the current response. Based on the expression of Equation (1), the changes in the impedance magnitude (denoted as  |Z|) of blood during coagulation could be estimated. The sinusoidal-waveform voltage was set to a 0.05 V amplitude and a 10 kHz frequency, considered as the best parameters to clearly identify the markers of coagulation. The sampling rate was chosen equal to one value per second (1 Hz), which is adequate to detect clotting variations that are of the minute order. The test was carried out in a small, closed chamber 0.5 cm^3^ in volume so as to prevent blood from drying. The measurement was performed in incubation at 37 °C for 45 min (Figure 1a,b). The recorded data were normalized to the first value (denoted as $\left|Z\right|/|Z_{t=0}|)$ and plotted as a function of time.

### 2.4. Impedance Phase Angle

The changes in the blood impedance phase angle (*Φ*) during coagulation were recorded using a Solartron spectrometer (1296A Dielectric interface System) at a 10 kHz frequency and a 0.05 V amplitude. Following the formation of the coagulum, retraction occurred, with the consequent expulsion of serum from the clot (Figure 2d). Serum was then withdrawn and separated from the clot, and the impedance phase of the clot and ejected serum were separately measured using broadband spectroscopy at different frequencies (from 10 Hz to 1 MHz) using the sample holder shown in Figure 1a.

### 2.5. Confocal Microscopy

A confocal microscope (Nikon A1r) was used to observe blood during coagulation to verify the significance of the markers highlighted in the impedance-magnitude test. A solution with sodium bicarbonate (0.1 M; pH 8.3) and Alexa Fluor™ 488 Conjugate was prepared (40 µg/mL), and 5 µL of it was added to 200 µL of citrated blood. The mixture was then incubated at 37 °C for 10 min, followed by the addition of calcium solution in the same ratio explained in Section 2.2. A stopwatch was set as soon as calcium solution was introduced, allowing us to control the delay between the addition of the coagulant and the beginning of the observation. A volume of 80 µL of blood was deposited on a closable glass-bottom dish (purchased from Mattek; No. coverslip 1.5; standard size of 35 mm diameter and 14 mm height; uncoated glass; sterilized with gamma-irradiation), to prevent blood from drying, and observation started in a closed chamber at a controlled temperature of 37 °C. Three different planes were appropriately selected to observe the coagulation process, at 0 µm, 3.28 µm and 6.57 µm from the glass. Every 27.3 s, an image per plane was recorded over 45 min. Only the pictures from the first plane (i.e., the most representative, 0 µm) are reported. The objective magnification was 40×, and the numerical aperture (NA) was 1.30. The emission wavelength was set to 525 nm with an excitation wavelength of 488 nm. Other setting parameters were chosen as follows: image pixel size of 0.31 µm/pixel, pinhole of 80 µm, optical sectioning of 1.5 µm and optical resolution of 0.21 µm. The same blood observed via microscopy was then analyzed using impedance spectroscopy.

### 2.6. ROTEM^®^ Characterization

To further validate the technique developed in this study, an impedance-magnitude analysis and an ROTEM^®^ test (using TOTEM delta Werfen) were simultaneously performed. Blood collected from the same donor was used during both tests performed at 37 °C. Nine tests were carried out on blood recalcified with calcium solution. For each of the nine tests, blood was drawn from a different donor. Subsequently, the extrinsic coagulation pathway was verified by performing six further tests with recalcified blood to which TF solution was added. The clotting time measured with ROTEM^®^ was then compared to the markers identified with impedance spectroscopy. A blood volume of 200 µL was used for the ROTEM^®^ analysis, and the ratio of blood to clotting agents was the same as mentioned in Section 2.2.

## 3. Results and Discussion

### 3.1. Impedance Spectroscopy

The impedance characterization of blood during coagulation allowed us to identify the main biomarkers of the key events. First, the impedance magnitude of citrated whole blood (without coagulants) was measured at a constant frequency of 10 kHz for 45 min. The same measurement was then repeated on recalcified blood, i.e., coagulating blood. Figure 2a illustrates the time evolution of the electric impedance of blood samples normalized to the first value ($\left|Z\right|/|Z_{t=0}|$). Substantial discrepancies between whole blood and clotting blood were observed, confirming that blood impedance spectroscopy can potentially define biomarkers to assess blood coagulation. The red curve of whole blood was characterized by an initial slight decrease, probably related to the formation of rouleaux of red blood cells. A few minutes after, an increasing trend associated with the sedimentation of red cells on the bottom appeared, leading to a phase separation of red cells and plasma. On the other hand, the black curve associated with clotting blood clearly presented a peak at around 9 min (the 1st marker) and a subsequent decrease culminating in a minimum at 13 min (the 2nd marker), beyond which a sudden steep increase appeared. Given the timing of the two markers, it is conceivable that they are related to fibrin formation and clot retraction, respectively. 

Figure 2d provides a sketch of the steps occurring in clotting blood during the measurement. A first verification of the second marker was achieved by comparing the |*Z*| measurements with a visual inspection of the clot. The retraction of the clot was observable with the naked eye in the sample, as shown in Figure 2b. If contraction did not occur (or was delayed beyond 45 min), no separation of phases was visible in the sample. As a result, the impedance curve did not manifest any increase but ended with a plateau indicating non-retracting blood (Figure 2c). This was a first verification of the meaning of the sharp increase in impedance, which was caused by a separation of the clot and serum phases due to the contraction of the thrombus. As reported in Figure 3, a similar trend was obtained for two impedance measurements performed on the same sample 24 h and 48 h after collection, confirming no significant changes in coagulation behavior. The time since withdrawal may have affected the platelet count; however, this effect did not impact clotting time nor clot retraction. 

A further assessment of the second marker was performed by measuring the impedance phase angles (*Φ*) using a Solartron analyzer system. Usually, the value of *Φ* indicates the capacitive, resistive or inductive behavior of a sample at a specific frequency. First, *Φ* at 10 kHz was measured in blood during coagulation to determine its trend over time. Then, clot and serum, separated after retraction, were tested using the broadband spectroscopy of *Φ* over a large frequency range from 100 Hz to 1 MHz. The variation in *Φ_Blood_* during coagulation followed the trend of normalized impedance magnitude, presenting the two main markers (Figure 4b). Regarding the rise that occurred after 13 min, the angle increased from −7.3° to −5.6°, indicating an increase in resistive behavior. According to the broadband-spectroscopy analysis, at 10 kHz, the clot had *Φ_Clot_* equal to −5°, which indicated highly resistive behavior (Figure 4c). Indeed, at moderate frequencies (~ from a few Hz to kHz), cells give a resistive response, as the electric field cannot penetrate the cell membrane acting as an insulating barrier, which is capacitive and only detectable in the range of MHz (Figure 4a) [36]. Serum broadband spectroscopy revealed a *Φ_Serum_* angle of −24°, denoting a less resistive behavior than that of the blood clot (Figure 4c). It is, therefore, intuitive that the increase in resistive behavior during coagulation was related to a contraction of the clot with the expulsion of the less resistive serum (Figure 2d). Finally, only the clot made of fibrin, platelets and red blood cells remained between the electrodes. Further investigations were carried out and are detailed in the paragraphs below to validate the first biomarker as well as to corroborate the second one.

### 3.2. Confocal Microscopy

The confocal microscopy of clots and coagulating blood was already performed in several investigations. The ability to track blood-related events in real time is undoubtedly of high interest for the condition monitoring of coagulation. For instance, Wallner et al. investigated how temperature affected platelet and coagulation activities as seen using real-time live confocal imaging [37]. A similar study reported by Kim et al. is equally noteworthy [38]. These findings indicated that the approach based on confocal imaging is efficient in the successful evaluation of clotting time together with clot retraction. The course of coagulation was imaged using confocal microscopy in order to further verify the markers found using impedance-magnitude spectroscopy. Fluorescent labeling emitting green fluorescence was added to blood to visualize the filaments of fibrin, otherwise invisible. As explained in Section 2.5, the first image was not pictured immediately after the addition of coagulant solution but rather 4 min and 40 s after recalcification, due to the time required to set the microscope. The initial image presented a green background with darker spots related to red blood cells, as indicated by the red arrow in Figure 5a, not visible in the analyzed wavelength range, while fibrin was still not present, since no polymerization had yet occurred (Figure 5a). When fibrinogenesis was activated, the first fibrin filament appeared 6 min after recalcification (Figure 5b). Fibrin began to form from a localized area and propagated in space until it covered the entire examined area, 11 min after the addition of calcium solution (Figure 5c) (Appendix A, which was speeded up by 64 times with respect to the original recorded in a real-time). The appearance of fibrin indicates the clotting time (CT). A contraction of the fibrin network could be noticed in the video and in the sequence of images reported below, starting from 11–12 min (Figure 5c,d). Platelets, identified as small circles at the vertices of the filaments, locally pulled fibrin causing whole-clot contraction at the macroscopic level (Figure 5d–f). To quantify this phenomenon, a reduced area of the whole image, circled in red in Figure 5c, was analyzed. The evolution of this area as coagulation proceeded is illustrated in Figure 6a–f, where the length of the filament was determined in pixels, using ImageJ^®^. Before 12 min, no reductions in the length of fibrin were measured. At minute 11, the filament shortened by 10% of its initial length, indicating the starting of retraction (Figure 6c), while at the end of the measurement, the length reduction was 55%. Following a confocal analysis, a blood sample from the same donor was tested using impedance-magnitude analysis, and the markers were compared with the confocal-microscopy observation. The clotting-time peak (the 1st marker) and the minimum followed by a steep increase (the 2nd marker) coincided exactly with the times detected using microscopy for CT and clot retraction, confirming what predicted in Section 3.1 (Figure 7). 

### 3.3. ROTEM^®^ Characterization

Characterization by means of impedance magnitude was compared with the ROTEM^®^ test, frequently used in biological laboratories as the gold standard for characterizing coagulation. Such a measurement quantifies viscoelastic changes in whole blood subjected to coagulation activation. Blood samples were simultaneously tested with the two techniques, and the acquired markers were compared. 

In a first set of tests (nine different blood samples), coagulation was only triggered by means of recalcification. Figure 8a displays the coagulation plot obtained with the impedance measurement of a blood sample, while Figure 8b illustrates the results collected with ROTEM^®^ for a sample from the same donor. As expected, the clotting time acquired during the impedance test (denoted as CT_Impedance_) was well correlated to the ROTEM^®^ one (denoted as CT_Rotem_), allowing us to clarify the meaning of the first peak in the impedance plot. The boxplot in Figure 9 provides a statistical overview of the results acquired with the nine tests performed on different donors’ blood, comprising the locality, dispersion and skewness of the acquired data. The numerical values are reported in Table 1, where mean value, standard deviation (SD), and maximum and minimum values are reported. The three quartiles (*Q*_1_, *Q*_2_ or median, and *Q*_3_) are also reported, where *Q*_1_ is the 25th percentile, i.e., the median of the lower half of the dataset; *Q*_2_ is the 50th percentile, i.e., the median of the entire dataset; *Q*_3_ is the 75th percentile, meaning, the median of the upper half of the dataset; and *IQR* is the interquartile range. To better assess the variability of the data, we here provide an estimation of the quartile coefficient of dispersion (*QCD*) given by Equation (3):(3)$$QCD=\frac{Q_{3}-Q_{1}}{Q_{3}+Q_{1}}=\frac{IQR}{Q_{3}+Q_{1}}$$

Overall, the values from ROTEM^®^ and the impedance spectroscopy matched. The average discrepancy between CT_Rotem_ and CT_Impedance_ was found to be around 11 s (2%), reflecting the high reliability of the impedance analysis that somehow agreed with the standard measurement. Based on ROTEM^®^ characterization, it was not possible, however, to verify the second impedance marker appearing at the beginning of clot retraction, since this value was not accessible using the viscosity test. As a matter of fact, ROTEM^®^ allows one to identify the maximum clot firmness (MCF), which indicates the maximum strength of the clot occurring after the retraction has already begun. 

To analyze coagulation via the extrinsic pathway, a second set of tests was performed by initiating coagulation with the addition of calcium and TF solutions. Six different blood samples were simultaneously characterized using ROTEM^®^ and impedance-magnitude tests. Impedance characterization enabled us to identify clotting time and retraction markers. Figure 8c,d report the results from the |*Z*| analysis and ROTEM^®^ test, respectively, of blood collected from the same donor. Good coherence between CT_Rotem_ and CT_Impedance_ was observed again, confirming the high accuracy of the developed protocol. Figure 9 displays the whisker box graph computed over all the tests. To some extent, the clotting times measured through these two techniques were analogous. Focusing on the average value, it was 3 min for both the impedance and ROTEM^®^ tests. 

Table 1 reports the detailed data related to the box graph, giving an overview of the statistics of the experiments. The results reported in Table 1 led to the following conclusions:Whatever the technique of measurement, recalcified blood samples with TF led to smaller values of CT;For all samples with or without TF, both the impedance and ROTEM^®^ techniques led to similar results in terms of central tendency (mean and median values), dispersion (SD and *IQR*) and limit values (max and min);In any case, the data distributions were almost symmetrical, as the medians (horizonal lines in the whisker box) were close to the mean values (circles); thus, the skewness should be near zero;None of the observations showed outliers or extremes values (i.e., no values fell below Q1 − 1.5 *IQR* or were above Q3 + 1.5 *IQR*), meaning that the highest and lowest occurring values were within this limit interval;Finner analysis regarding the data dispersion via *QCD*:
✓The impedance technique gave rise to a somewhat higher dispersion of CT than the ROTEM^®^ one; ✓The measurement of the samples without TF resulted in lower dispersion, regardless of which technique was chosen;✓For all the cases, the *QCD* value was relatively low, confirming the good repeatability of the data.

Accordingly, the comparison between ROTEM^®^ and the impedance-magnitude analysis confirmed the significance of the 1st biomarker in recalcified blood, either with or without TF. 

The values of CT achieved in this work (from 6.5 to 10 min) are similar to those reported in other studies. Vrigkou et al. found an average of 8.1 min in healthy patients, confirming an increase in CT in patients with chronic thromboembolic pulmonary hypertension [39]. Similar findings were discovered by Sucker et al.; the clotting time activated using recalcification in women equaled 7.4 min, while this was 8.6 min in men [40]. In addition, Ramström et al. found a range of CT values similar to the one reported in our work [41]. In other words, the procedure defined in this study in terms of storage time and temperature did not seem to significantly affect the coagulation time. 

## 4. Conclusions

This work reports on innovative biomarkers for the assessment of blood coagulation based on impedance spectroscopy characterization. Such an approach provides a real-time and comprehensive picture of all the events occurring during the clotting of blood. Two significative markers are identified, i.e., markers related to fibrin formation (the 1st marker) and clot contraction (the 2nd marker). The impedance phase (*Φ*) investigation confirms the meaning of the 2nd marker, revealing an increase in the resistive behavior during contraction caused by the expulsion of serum. Additionally, imaging performed with confocal microscopy provides the onset time of fibrin polymerization as well as the beginning of clot contraction, enabling the physical meaning of both test markers to be further highlighted. In addition, |*Z*| characterization compared with the standard ROTEM^®^ analysis allows the meaning of the 1st marker to be strengthened. The impedance-magnitude test gives an overview of the clotting time, which could be usefully used as an effective indicator of hypo-/hyper-coagulability disorders. The test proposed in this study is not only reliable in terms of identifying clotting time, but it also allows one to detect the onset of clot retraction. Indeed, if contraction does not occur or is delayed, the test should not show the 2nd marker followed by a steep increase. This technology, relying on the electrical-impedance characterization of blood, allows the real-time monitoring of coagulation to be conducted, which is otherwise inaccessible with current techniques (ROTEM^®^ analysis). The analyses of the effect of external factors (temperature, time after collection, diseases) on coagulation are outside of the scope of this paper; nonetheless, they should be among the highest priorities in our future research direction. Indeed, the approach used in this study involves testing blood 24 h after collection; however, it is not fully disclosed how much storage time may alter clotting time. In any event, the findings of this study are consistent with those of other studies conducted with ROTEM using the same coagulants, suggesting that testing within 24 h does not significantly alter the results. However, further investigations are still required to better confirm this matter. A novel insight of this work involves the development of an easy and efficient technique that might be adaptable to other materials’ surfaces to analyze their interactions with blood. The proposed approach enables the determination of the thrombogenicity of materials to be performed in order to identify the most hemocompatible candidates. A study of materials’ thrombogenicity is crucial because one of the main limitations of cardiovascular prostheses is their tendency to induce thrombosis once in contact with blood. Hence, there is an urgent need for materials capable of delaying or avoiding coagulation to prevent patients from severe health problems such as ischemia, heart attack or embolism. Impedance blood characterization may, therefore, pave the way for a new generation of smart materials that could overcome the current technological lock of cardiovascular prostheses for which coagulation is a serious issue. 

## Figures and Tables

**Figure 1 biomedicines-10-01833-f001:**
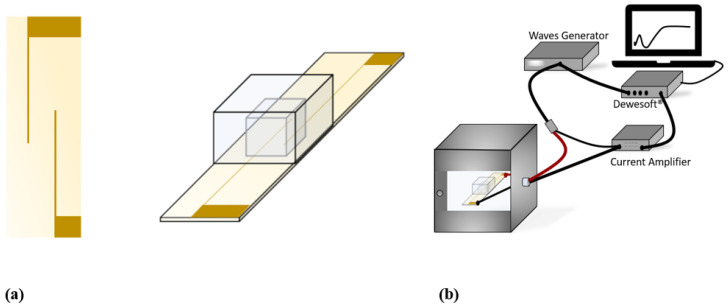
(**a**) Sample holder used for blood impedance-magnitude spectroscopy. (**b**) Complete set-up employed in |*Z*| measurements.

**Figure 2 biomedicines-10-01833-f002:**
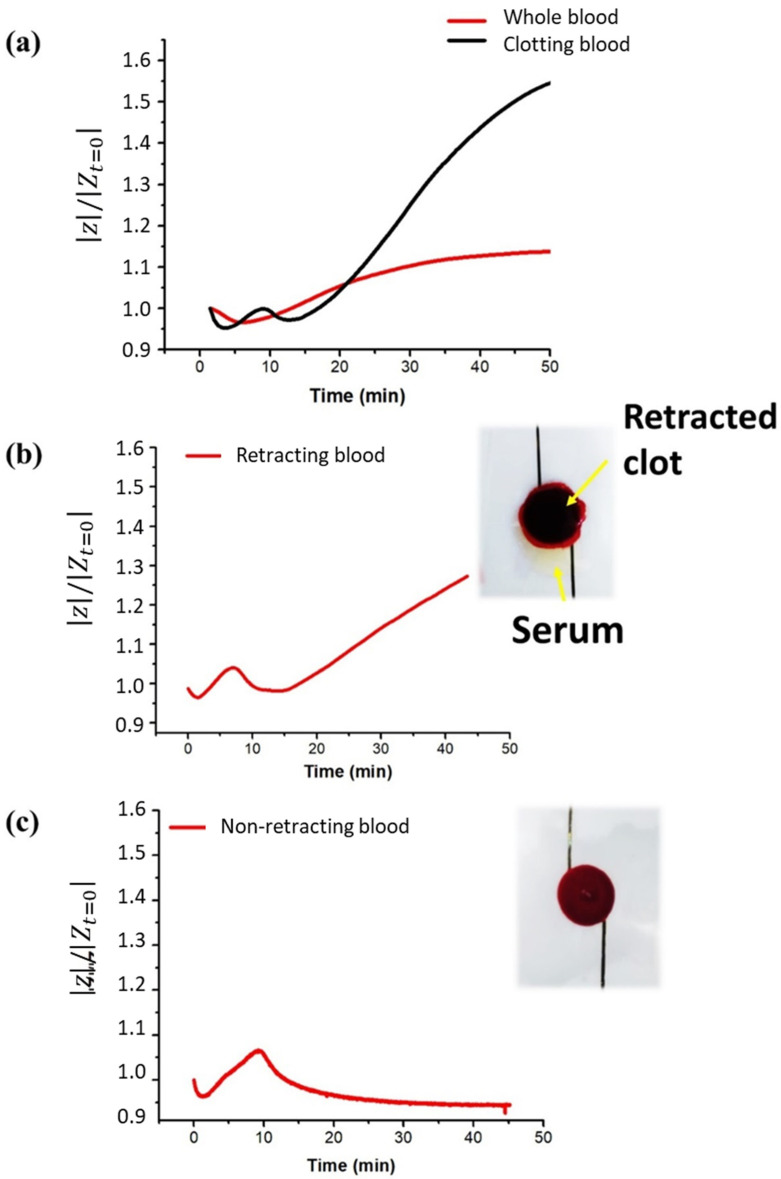
Time evolution of normalized impedance |*Z*| measured at 10 kHz for: (**a**) whole blood and recalcified blood; (**b**) retracting blood with respective picture of retracted blood clot; and (**c**) non-retracting blood with picture of non-retracted clot. (**d**) Sketch illustrating the steps occurring in clotting blood during the impedance-magnitude test.

**Figure 3 biomedicines-10-01833-f003:**
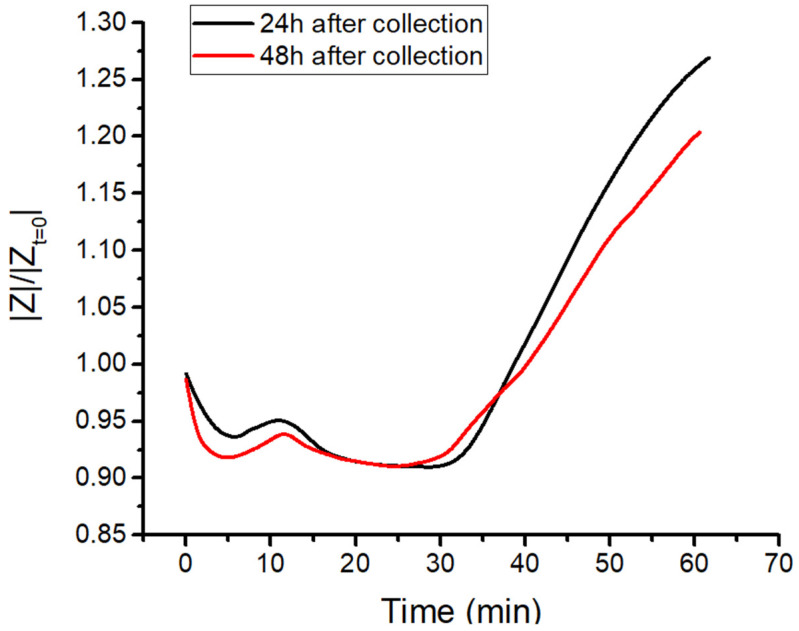
Impedance test of blood coagulation of the same donor’s sample tested 24 h (black curve) and 48 h (red curve) after collection.

**Figure 4 biomedicines-10-01833-f004:**
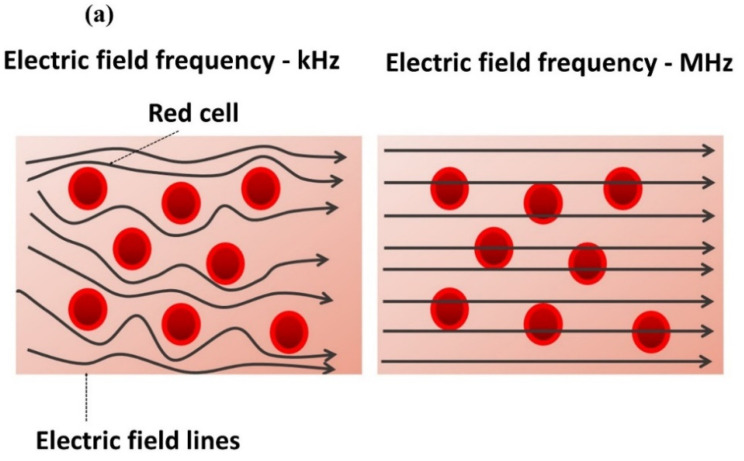
(**a**) Electric-field lines in blood at low frequency (kHz) and high frequency (MHz). (**b**) Impedance phase angle (*Φ*) as a function of time for recalcified blood; the red arrows indicate the value of *Φ* before and after the steep increase. (**c**) Impedance-phase broadband spectroscopy (100 Hz–1 MHz) of clot and serum separated after retraction.

**Figure 5 biomedicines-10-01833-f005:**
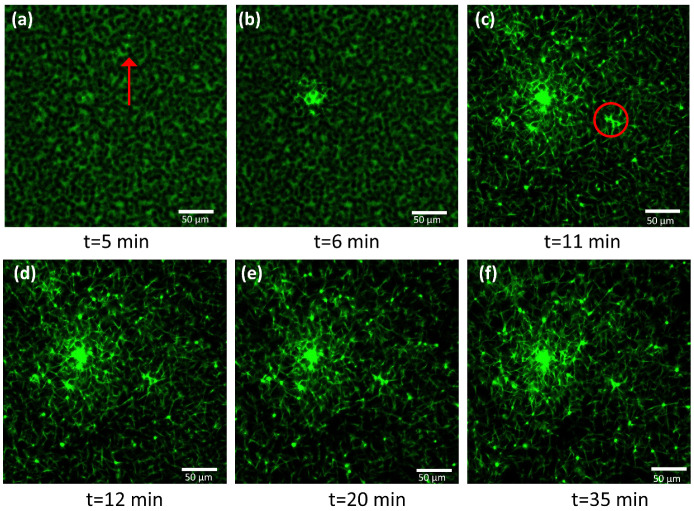
(**a**–**f**) Confocal-microscopy frames at different times since the addition of calcium solution. The red arrow in figure (**a**) indicates the dark shadow of a red cell. The green fibers correspond to the fibrin network labeled using fibrinogen from human plasma Alexa Fluor™ 488 conjugate.

**Figure 6 biomedicines-10-01833-f006:**
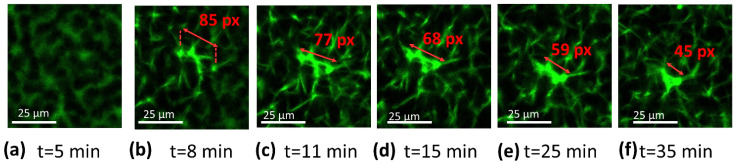
(**a**–**f**) Detail of a fibrin branch from its appearance at different moments of the contraction. In each frame, the length of fibrin was measured in pixels using ImageJ^®^ software.

**Figure 7 biomedicines-10-01833-f007:**
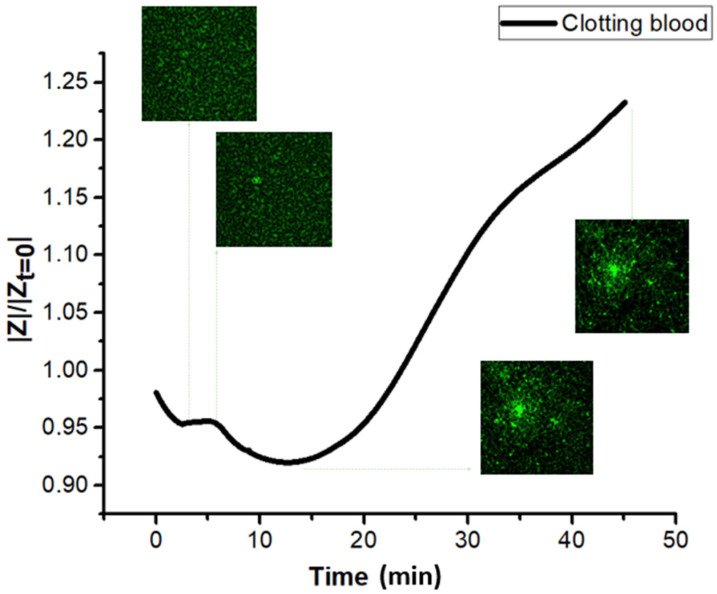
Impedance-magnitude test of clotting blood from the same donor of that tested with confocal microscopy.

**Figure 8 biomedicines-10-01833-f008:**
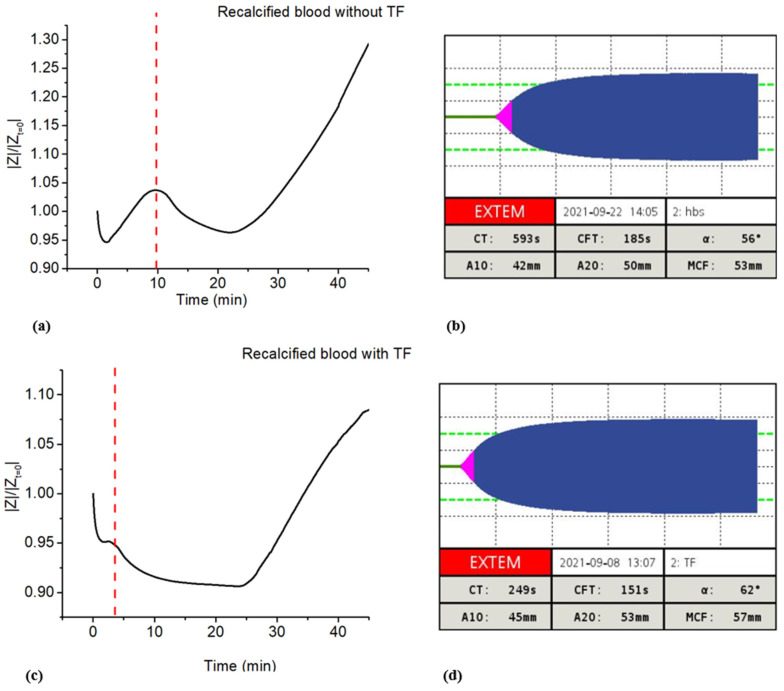
(**a**) Impedance-magnitude test and (**b**) ROTEM^®^ analysis of recalcified blood from the same donor without TF. (**c**) Impedance-magnitude test and (**d**) ROTEM^®^ analysis of recalcified blood from the same donor with TF.

**Figure 9 biomedicines-10-01833-f009:**
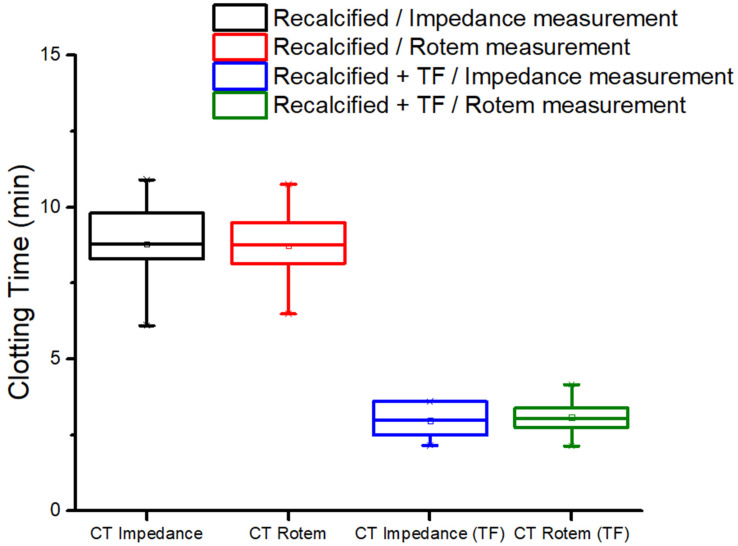
Box graph computed over 9 different blood samples (recalcified without TF) representing clotting time measured using impedance-magnitude spectroscopy and ROTEM^®^ (black and red boxes, respectively). The blue and green boxes represent CT measured using impedance-magnitude spectroscopy and ROTEM^®^ analysis, respectively, on 6 different samples of recalcified blood with TF.

**Table 1 biomedicines-10-01833-t001:** Statical data related to CT_Impedance_ and CT_Rotem_ for recalcified samples with and without TF.

Variable: CT	Impedance		ROTEM^®^	
	With TF	Without TF	With TF	Without TF
Mean value (min)	2.96	8.76	3.08	8.71
SD (min)	0.65	1.38	0.68	1.23
Min value (min)	2.15	6.10	2.13	6.46
Max value (min)	3.60	10.90	4.15	10.75
*Q*_1_ (min)	2.50	8.30	2.73	8.13
*Q*_2_ (min)	2.99	8.80	3.04	8.76
*Q*_3_ (min)	3.60	9.80	3.41	9.50
*IQR* (min)	1.10	1.50	0.68	1.36
*QCD* (%)	18.0	8.3	11.1	7.7

## Data Availability

Not applicable.

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
