# Peer review of "Investigation of Blood Coagulation Using Impedance Spectroscopy: Toward Innovative Biomarkers to Assess Fibrinogenesis and Clot Retraction"

_biomedicines, 2022, doi:10.3390/biomedicines10081833_

Round 1
Reviewer 1 Report
In the presented resubmitted manuscript, the authors intend to demonstrate innovative biomarkers for assessment of blood coagulation. In general, an extensive study was made and some corrections however there are still several issues regarding the presented research.
1. The authors used blood samples stored over 24h after collection which seems too long regarding platelet survival and function and stability of coagulation factors. The authors explained it was due to the procedure in their Local Blood Bank and tried to prove that all the coagulation was stable. So in the Fig. 3 it was shown that the blood coagulation was similar between 24 h and 48h. Unfortunately, such a data is not convincing since if we know that coagulation is stable up to 6-8h so the comparison between 24 and 48h does not prove the stability of the coagulation. The only solution in this case is to compare 2h and 24h.
Moreover, to prove the stability of the blood coagulation after such a long time (24h) the authors cited two papers which is reference no 33 and 34. However the cited papers did not answer the issue.
In the cited paper 34 (Toulon et al. 2017) The authors conclude: ‘PT/INR, aPTT, fibrinogen, FV, and D-dimer can be reliably evaluated in tubes stored unspun at room temperature for up to 8 hours after blood collection. That optimal delay should be of 6 hours for FVIII.’ So in this paper there is no data on the stability of coagulation after 24h there.
In the second cited paper 33 (Denessen et al. 2022) the authors concluded; ‘ Our data indicated that, for the PFA, whole blood can be stored for 3 hours at room temperature. Whole blood used for the Multiplate and ROTEM can be stored for 6 hours of storage. For LTA, PRP and whole blood were stable up to 3 hours at 4°C or room temperature and 6 hours at room temperature, respectively.’
So unfortunately, the authors of the presented manuscript did not provide any proof or references to confirm the blood samples stored for so long are still reliable and coagulation is fully active.
Please, make a comment.
2. In some parts English language correction is required: like in Line 42 ‘….The clot and plugs the wound to minimize….’
3. In the section results and discussion the authors a bit discuss their results but the so no much discussion with other researchers.
4. How is it clinically relevant the data?
Reviewer 2 Report
Thank you for considering the suggestions from my first review report. I think the manuscript has improved a lot. In my opinion, the key takeaways are now much clearer for the reader. I have no further change requests.
Author Response
We thank to the Reviewer for having appreciated our investigations and for the valuable comment have been raised in the last review process.
Round 2
Reviewer 1 Report
ok
This manuscript is a resubmission of an earlier submission. The following is a list of the peer review reports and author responses from that submission.
Round 1
Reviewer 1 Report
In the presented manuscript, the authors intend to demonstrate innovative biomarkers for assessment of blood coagulation. In general, an extensive study was made however there are several issues regarding the presented research.
1. According to definition: biomarker is a biological molecule found in blood, other body fluids, or tissues that is a sign of a normal or abnormal process, or of a condition or disease.
What exactly is the novel biomarker in the presented manuscript?
2. The authors used blood samples stored 48h after collection. So there is a question about the quality of the research. It is commonly known that platelet function must be assessed up to 3-4h after blood collection. Moreover, blood platelets in a buffer without glucose are not able to survive for long hours. Additionally coagulation factors are stable 3-5 hours in the whole blood and for any routine analysis the citrated samples are frozen down shortly after blood collection if the coagulation factors are not measured within a few hours.
Please make a comment.
3. In the introduction there are several issues regarding English writing and use of words. For example; we use term: coagulation factors, clotting factors but not coagulant factors.
The term maturation for the …fibrin network (line 36) is not correct. It is better to use: production, activation, triggering.
4. A few issues regarding incorrect physiological terms: line 46- thrombocytopenia is not a diseases and it is not a disorder of platelet function but it is related to low platelet count.
Tissue factor and collagen is not in the endothelium but in the subendthotelium which is exposed due to vessel injuries.
5. There is no need add paragraph 2 in the introduction (line 86) and fig 1. Coagulation cascade is commonly and widely known and well explained by many other authors. Additionally, your fig 1 is not your original date, such a figure can be find elsewhere.
Also, what is 1ry or 2ry in the figure 1?
6. In methods- what was the final concentration of Cacl2 recalcification?
7. In fogures 5 and 6. There is no need to show all the pictures, show just a few showing real diffrences.
8. There is discussion missing in the manuscript. Please discuss with results of other researchers.
Reviewer 2 Report
The manuscript "Investigation of blood coagulation via impedance spectroscopy: Toward innovative biomarkers to assess fibrinogenesis and clot retraction" is well written. The authors describe an interesting approach of detecting / monitoring blood coagulation with impedance spectroscopy. They show the accordance of the method to ROTEM and microscopy to point out the accuracy of the method and mention the advantages. Description of the methodology and background information are concise.
One suggestion: The authors should more clearly point out the advantages of the new method compared to the current ones. There is some description in the paragraph lines 70-85, but in the conclusion it could be more emphasized, especially for the field of materials’ characterization for cardiovascular prostheses. Maybe there is a "unique selling point" that could be highlighted.
As multiple measurements are mentioned in the text, some statistics would be nice to underline the reproducibility of results and possible variations.
The authors nicely show the fibrin network with confocal microscopy. For a better idea of size, scale bars would be helpful in all the microscopy images.
Some typing errors:
Line156: could beestimated - a space is missing
Line 315: is well correlated to the ...
Line 320: reliability to the spectroscopic analysis ...
Authors: G. D’Ambrogio1 - number instead of superscript